# Recent Trends in Oceanic Conditions in the Western Part of East/Japan Sea: An Analysis of Climate Regime Shift That Occurred after the Late 1990s

Hae-Kun Jung [1],* , S. M. Mustafizur Rahman [2] , Hee-Chan Choi [1], Joo-Myun Park [3] and Chung-Il Lee [4]

[1] Fisheries Resources and Environment Research Division, East Sea Fisheries Research Institute, National Institute of Fisheries Science, Gangneung 25435, Korea; gmlckschl82@korea.kr
[2] Department of Oceanography and Hydrography, Faculty of Earth and Ocean Science, Bangabandhu Sheikh Mujibur Rahman Maritime University, Mirpur, Dhaka 1216, Bangladesh; mustafizur.ocn@bsmrmu.edu.bd
[3] Dokdo Research Center, East Sea Research Institute, Korea Institute of Ocean Science & Technology, Uljin 36315, Korea; joomyun.park@kiost.ac.kr
[4] Department of Marine Bioscience, Gangneung-Wonju National University, Gangneung 25457, Korea; leeci@gwnu.ac.kr
* Correspondence: hkjung85@korea.kr; Tel.: +82-33-660-8533

**Abstract:** The western part of East/Japan Sea (WES) is an important area for understanding climate change processes and interactions between atmospheric and oceanic conditions. We analyzed the trends in recent oceanic conditions in the WES after the recent climate regime shift (CRS) that occurred in the late 1990s in the North Pacific. We explored the most important climate factors that affect oceanic conditions and determined their responses to changes in climate change. In the CRS that occurred in the late 1980s, changes in oceanic conditions in the WES were influenced by intensity changes in climate factors, and, in the late 1990s, it was by spatial changes in climate factors. The latitudinal shift of the Aleutian low (AL) pressure influences recent changes in oceanic and atmospheric conditions in the WES. The intensity of the Kuroshio Current and the sea level pressure in the Kuroshio extension region associated with the latitudinal shift of the AL pressure affects the volume of transport of the warm and saline water mass that flows into the WES and its atmospheric conditions. In addition, the fluctuations in the oceanic conditions of the WES affect various regions and depth layers differently, and these variations are evident even within the WES.

**Keywords:** western part of East/Japan Sea; climate regime shift; Kuroshio extension; Aleutian low pressure; pressure gradient





## 1. Introduction

The western part of East/Japan Sea (WES, also known as Sea of Japan) is geographically located in the Northwest Pacific Ocean, on the boundary between the North Pacific Ocean (NPO) and the Asian continent. Because cold and warm currents coexist in the WES, a thermal front is clearly formed at the boundary between the cold and warm regions [1,2]; additionally, physical environmental characteristics, such as eddies and deep ocean circulation, occurring in large oceans are formed [3]. In addition, the WES is located at the boundary between oceanic air masses and continental air masses; therefore, the atmospheric conditions of the WES are affected by various atmospheric pressure systems simultaneously [4–7]. As a result, seasonal and inter-annual changes are clearly observed in atmospheric conditions, such as wind and air temperature [8–10].

The oceanographic conditions in the WES are related to the oceanic and atmospheric circulation systems in the NPO. In particular, climate regime shifts (CRSs) that occurred in the North Pacific in the mid-1970s, late 1980s, and late 1990s caused a change in the thermal energy flowing into the WES through the oceanic and atmospheric circulation systems in the NPO [5,7]. Consequently, the oceanic conditions in the WES repeatedly fluctuated

by crossing the warm and cold regimes at a time similar to the CRS [4,11,12]. Previous studies on the changes in oceanic conditions of the WES associated with the CRSs mainly focused on the intensity changes of climate factors to understand the mechanisms and relationship between CRS and WES oceanic conditions [4,5,7]. Especially, the relationship between oceanic conditions in the WES and the climate index, which indicates the oceanic and atmospheric conditions in the NPO, was analyzed [4,11,12]. However, climate change in the NPO was affected simultaneously by changes in the intensity and spatial change of the climate factor [13–16].

The Tsushima warm current (TWC) flowing into the East/Japan Sea transports thermal energy from the large ocean to the WES [17,18]. The TWC is separated from the Kuroshio Current, passes through the western part of Kyushu, Japan, and flows into the WES through the Korea Strait [2,19]. It is divided into two paths: one path passes through the eastern channel of the Korea Strait and the northern coast of Honshu, Japan, while the other path (East Korea Warm Current: EKWC) passes through the western channel of the Korea Strait [2,19]. Subsequently, EKWC meets the North Korea Cold Current flowing along the North Korean coast through Primorsky Krai and Vladivostok, forming a current front in the WES [20,21]. This volume transport of warm and saline water mass following the WES was affected by changes in oceanic and atmospheric circulation in the NPO [5,22]. In particular, changes in the intensity of the Aleutian low pressure (AL) and Pacific Decadal Oscillation (PDO) are related to the oceanic circulation system in the North Pacific, including the Kuroshio Current [22]. Notably, changes in the oceanic circulation system associated with CRS influence the change in the volume of transport of the warm and saline water mass separated from the Kuroshio Current that flows into the WES [11,23,24]. In the case of the strengthened volume of transport of warm and saline water mass in the WES, the central axis of the EKWC moves more toward the coastal area of the WES [11,25], resulting in a northward shift of the thermal front in the WES, along with the thickening and deepening of the warm-water layer [11,23,24].

The fluctuations in atmospheric conditions during winter around the WES are affected by the continental high pressure around Eurasia (Siberian High Pressure: SH) and oceanic low pressure (AL) around the Aleutian Islands in the North Pacific, simultaneously [5,8,26]. As a result, strong northwest and northeast winds flowed into the WES, owing to the pressure gradient between the SH and AL [5]. The winter monsoon system around Korea serves to supply cold and dry air masses formed in the Siberian region into Korea, and acts as a major factor influencing the change in atmospheric conditions around the WES [4,5]. These atmospheric conditions around the WES exchange thermal energy with the ocean through friction with the sea surface and heat exchange, and these effects are transmitted to changes in oceanic conditions, such as water temperature [12,27], intensity of stratification [7], and mixed layer depth [3]. In particular, the changes in the intensity of atmospheric factors, such as SH and AL, according to the CRS, act as major factors that influence the changes in the oceanic conditions of the WES [5,7].

Notably, the oceanographic conditions in the WES have a distinct decadal cycle, and this cycle is closely related to CRS. In particular, during the CRSs in the mid-1970s, late 1980s, and late 1990s, changes in the thermal energy due to intensity changes in climate factors (representing the atmospheric and oceanic conditions in the North Pacific) acted as a major influencing factor of the changes in the WES oceanic conditions [4,11]. After the late 1970s, the intensity of AL strengthened, and the PDO showed a positive phase. As a result, oceanic conditions in the WES showed regime shifts from warm to cold conditions, with weakened volume transport of the warm water mass and cold atmospheric conditions around the WES [5,7,11]. However, after the late 1980s, the intensity of the SH and the East Asian Winter Monsoon (EAWM) weakened, and Arctic Oscillation (AO) showed abrupt changes from a negative to positive phase. As a result, the thermal energy flowing into the WES through the oceanic and atmospheric circulation system increased with the regime shift (from cold to warm conditions) in the WES [5,7,8]. After the 1990s, the frequency of regime shifts in the North Pacific gradually shortened, and the range of inter-annual

fluctuations was large [14,16]. In the late 1990s, unlike in the past CRS, spatial shift in the climate factors was more predominant than changes in their intensity [13–16]. As a result, the mechanisms and processes that occur in the recent years, causing spatial changes in climate factors, affect the changes in oceanic conditions that may be different from those that occur because of changes in intensity [28,29]. Thus, it is necessary to understand the causes and mechanisms of the recent changes in the oceanic conditions of the WES from the perspective of intensity and spatial changes in climate factors.

The aim of this study was to identify the following phenomena by analyzing recent changes in the oceanic conditions of the WES. (1) In this study, we analyzed, with respect to the recent CRS, the response of the oceanic conditions in the WES to the intensity and spatial distribution of climate factors. (2) We also analyzed the difference in the regional response of the WES with respect to the changes in the atmospheric and oceanic conditions around the North Pacific.

## 2. Data and Methods

### 2.1. Oceanographic Data

2.1.1. Water Temperature and Salinity in the Western Part of East/Japan Sea and East China Sea

In our study, oceanographic data, such as water temperature and salinity in the WES, were observed using conductivity, temperature, and depth (CTD) probe provided by the National Institute of Fisheries Science (NIFS). A total of 69 fixed stations were observed in February during 2016–2021. In this study, we focused on inter-annual changes in oceanic conditions of the upper layer (≤200 m), because changes in the oceanic conditions in the upper layer are more variable and respond more sensitively to atmospheric conditions than those in the bottom layer (Figure 1).

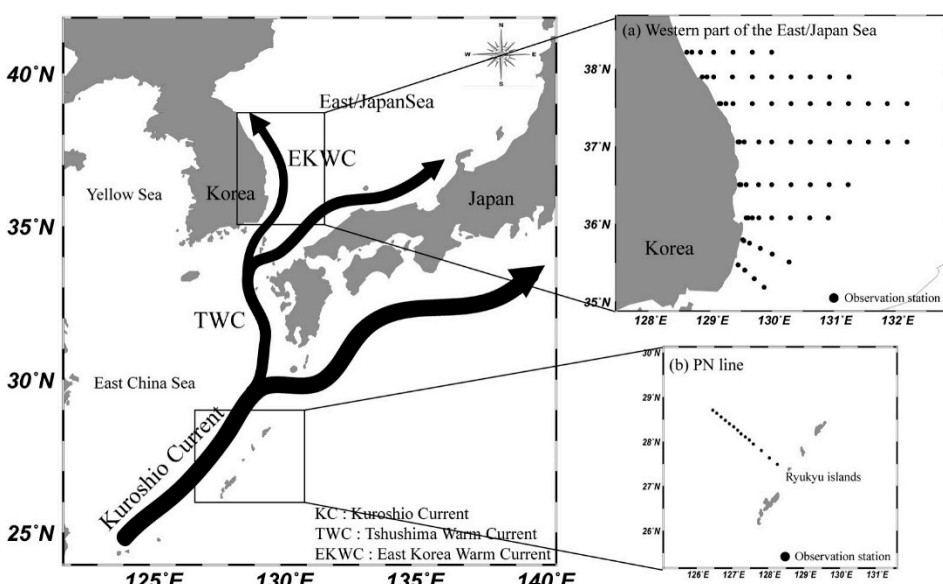

**Figure 1.** Study areas for sea water temperature and salinity analyses in (a) the western part of East/Japan Sea and (b) East China Sea (PN line).

To analyze inter-annual changes in the oceanographic conditions around the East China Sea (ECS) across the Kuroshio Current, oceanographic data at the PN section, which were observed on a quarterly basis using a CTD probe provided by the Japan Meteorological Agency (JMA) [30], were used (Figure 1). Inter-annual changes in the vertical structure of water temperature and salinity in January across the PN line for the period 2016–2019 were analyzed. Fourteen fixed stations from 126.45° E to 128.25° E along the PN line were observed vertically from 0 m to 1000 m, while in 2019, a total of seven stations were observed from 126.45° E to 127.16° E (Figure 1).

### 2.1.2. Determination of the Mixed Layer Depth (MLD)

In this study, we defined the mixed layer depth (MLD) as the depth at which the temperature is lower than that at the surface reference depth by a given threshold ($\Delta T$). This is one of the most popular methods because it can be applied to profiles with varying vertical resolutions [31]. To determine MLD with suppressed diurnal and retained long-term variabilities, a threshold of $\Delta T = 0.2$ °C and a reference depth of 10 m were used in this study [32–34]. The annual change in the MLD in the WES was analyzed by dividing it into inshore regions that were shallower than 500 m and offshore region where the bottom depth was deeper than 500 m. In addition, the fluctuation patterns in the MLD were analyzed, according to changes in the atmospheric and oceanic conditions.

### 2.1.3. Determination of Geographic Location and Vertical Volume of the East Korea Warm Current (EKWC)

The EKWC is generally northward along the Korean coast [35], but the main path and vertical volume can vary every year [25,36]. Therefore, the geographic location and vertical volume of the EKWC were estimated for each year in the time series. To calculate the location and vertical volume of the EKWC, we selected a region of 36.5° N or less, such that it had a relatively small frequency of regional-scale oceanographic phenomena, such as eddies [37]. The central location of the EKWC was calculated using an averaged latitude and longitude of the observation station indicating a reference temperature (>12 °C) and salinity (>34.3 psu) range at the depth of 100 m. The vertical length of the EKWC was estimated by the vertical length of the water mass, indicating the physical properties of the EKWC (temperature: $\geq 12$ °C, salinity: $\geq 34.3$ psu) [38].

### 2.1.4. Geostrophic Sea-Surface Current

To analyze the inter-annual changes in the horizontal distribution of the sea surface currents in the WES, we used the satellite geostrophic current data collected in February of every year during 2016–2021. The geostrophic sea surface current data were obtained from the Archiving, Validation, and Interpretation of Satellite Oceanographic (AVISO: https://www.aviso.altimetry.fr/en/home.html (accessed on 5 May 2021)) database archived monthly on a 0.25° grid from 2016 to 2021.

### 2.2. Atmospheric Conditions in the Western Part of East/Japan Sea (WES) and Climate Index

Inter-annual changes in atmospheric conditions such as wind speed and air temperature in the WES during winter (January and February) were analyzed using the ERA5 reanalysis data, which are the latest climate reanalysis data produced by the European Centre for Medium-Range Weather Forecasts (ECMWF) [39], along with the monthly data from 2016 to 2021 obtained at 0.25° × 0.25° spatial resolution. To analyze the relationship between the latitudinal shift of the AL and atmospheric conditions in the WES during winter (January and February), the latitudinal position time series of the AL was analyzed (Figure 2). We defined the latitudinal position of the AL using the sea level pressure (SLP) dataset at 0.25° × 0.25° spatial resolution by ERA5 from the ECMWF. The latitudinal position of AL was defined as the location of the minimum SLP value in latitudinal averaged SLP within a region of AL system (30–65° N, 160° E–140° W) [40]. The Siberian high-pressure index (SHI) [26], Aleutian low-pressure index (ALPI) [40], and East Asian winter monsoon index (EAWMI) [8] in winter (January and February) were used to describe the effect of atmospheric conditions in the North Pacific on the oceanic conditions in the WES. The pressure gradient between the SH-AL and SH-Kuroshio Extension (KE) region was defined by the SLP difference between the reference regions of the SH (40–60° N, 70–120° E), AL (30–65° N, 160° E–140° W), and KE (35–40° N, 140–150° E) (Figure 2).

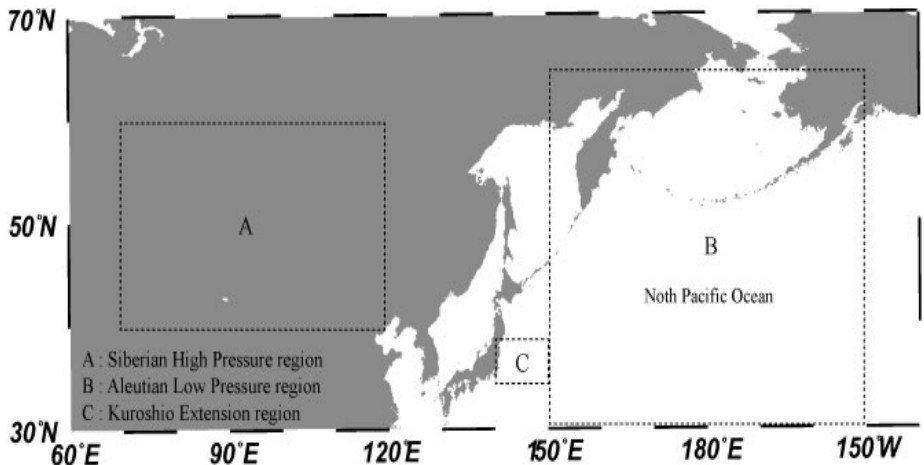

**Figure 2.** Sea level pressure regions: (A) Siberian high pressure, (B) Aleutian low pressure, and (C) Kuroshio extension region.

## 3. Results

### 3.1. Recent Oceanic Conditions in the Western Part of East/Japan Sea (WES): Water Temperature

In our study, the water temperature in the near surface (10 m) of the WES during winter was characterized by large inter-annual fluctuations. In particular, the range of the inter-annual fluctuation in the water temperature in the middle (36–37° N) and northern (≥37° N) regions was larger than that in the southern (<36° N) regions, where the annual water temperature was higher than 13 °C (Figure 3a). In 2016 and 2018, the water temperature of the near surface in the WES was 0.52 °C, which is lower than the mean water temperature calculated during 2016–2021 (i.e., 11.14 °C). Notably, the water temperatures in 2017, 2019, 2020, and 2021 were 0.7–1.9 °C higher than the mean water temperature estimated for 2016–2021. In particular, unlike other periods, in 2016 and 2018, the near surface with water temperatures below 10 °C were formed in the middle and northern regions of the WES (Figure 3a).

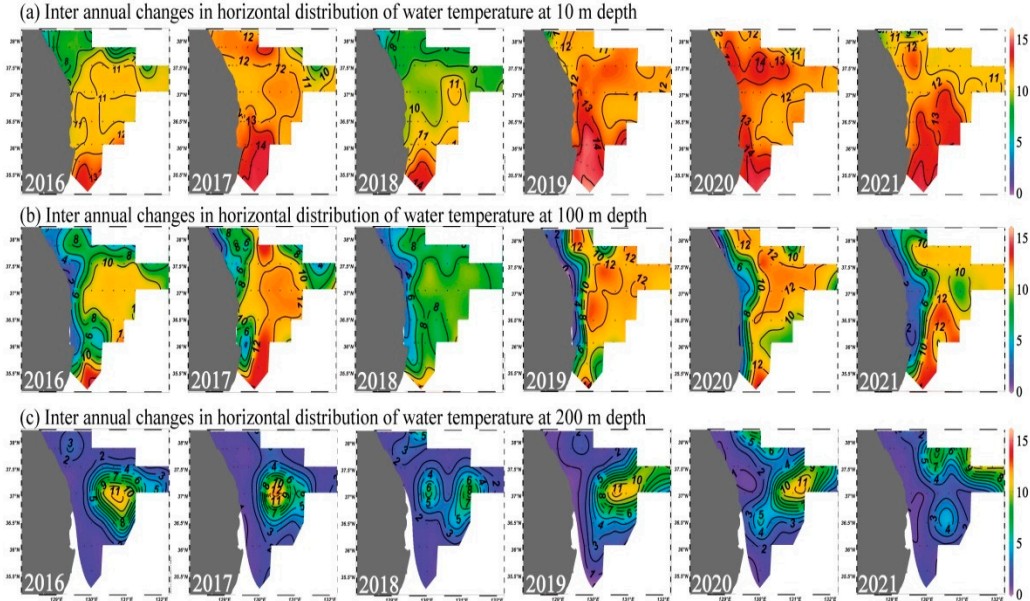

**Figure 3.** Inter-annual changes in horizontal distribution of water temperature (°C) at depths of (**a**) 10 m, (**b**) 100 m and (**c**) 200 m in the western part of East/Japan Sea (WES).

The annual change in water temperature at the depth of 100 m showed an inter-annual fluctuation pattern similar to that of the upper layer (Figure 3b). However, the water

temperature at the depth of 100 m in the coastal region showed an opposite trend to those occurring at 10 m and 50 m. Notably, in 2019, 2020, and 2021, the water temperature in the upper layer of the coastal region was higher (Figure 3b), but the water temperature at the depth of 100 m in the coastal region was lower during the same period, and the areas where the water temperature was 3 °C or lower were formed in coastal regions (Figure 3b). The annual change in water temperature at 200-m depth was different from that of the upper layer, and the water temperature stayed consistent below 2 °C in most areas annually (Figure 3c). Notably, the range of fluctuations in the water temperature annually was small compared to that in the upper layer (Figure 3c). However, a warm spot was formed where the water temperature was 8–9 °C higher than that of the surrounding region (Figure 3c). As a result, the water temperature at 200 m depth was mainly affected by the size and intensity of the warm spot and it continued to change annually.

### 3.2. Recent Oceanic Conditions in the Western Part of East/Japan Sea: Mixed Layer Depth

The winter MDL of the WES was deeper in the offshore (>500 m) region than in the inshore (≤500 m) region (Figure 4). From 2016 to 2021, the average of the MLD in the inshore and offshore regions were 39.7 m and 74.9 m, respectively (Figure 4). In the offshore region, a specific spot, with a much deeper MLD than that of the surrounding area, was formed. In particular, in 2016 and 2017, the formation of such a spot was more pronounced near 37° N (Figure 4). The formation of these deep MLD spots is related to changes in the surface currents. In the region where the MLD was deeper than the surrounding area, an anti-cyclone vortex (warm eddy) was formed (Figure 5). In particular, in 2016 and 2017, a strong warm eddy occurred near 37° N, and the MLD at the center of the warm eddy was significantly higher than that in the surrounding area (Figure 5).

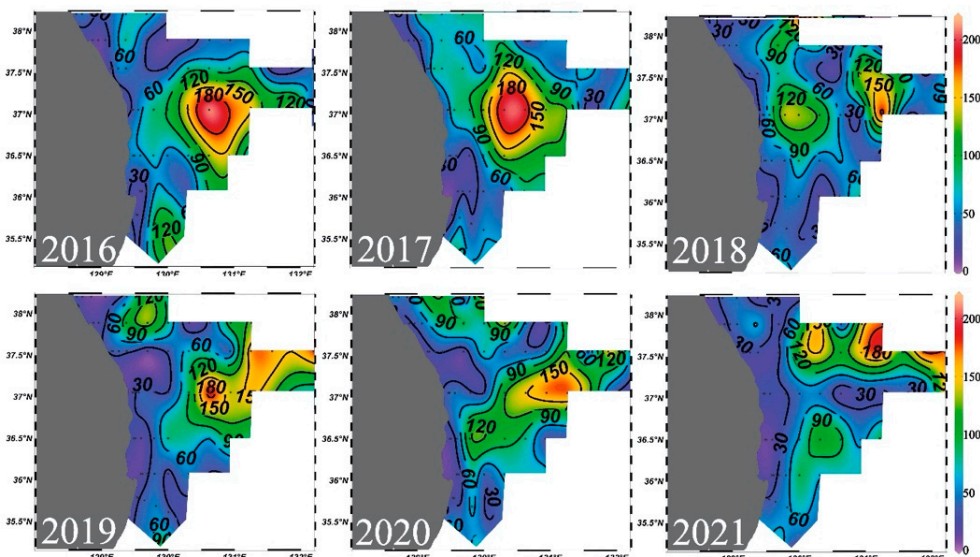

**Figure 4.** Inter-annual changes in horizontal distribution of mixed layer depth (MLD) (m) in the western part of East/Japan Sea (WES).

The annual changes in the MLD in the coastal region were the highest at 47.8 m in 2016, and they gradually decreased to the lowest at 32.3 m in 2019 (Figure 6a). These fluctuation patterns were related to atmospheric conditions such as wind speed (Figure 6a). The wind speed in the WES from 2016 to 2018, when the MLD was deep, was stronger than that in other periods; from 2019 to 2021, when the MLD was shallow, the wind speed in the WES was weak (Figure 6b). In addition, from 2019 to 2021, when the MLD was shallow, the water temperature in the upper layer continued to rise, and the water temperature at the depth of 100 m decreased in the opposite direction, forming a strong thermocline (Figure 6a,b). Notably, the annual change in MLD in the coastal zone is related to atmospheric factors,

such as wind speed, and the changes in the water column structure. However, the annual change in the offshore MLD has a greater effect on the change in the state of the oceanic current such as the formation of eddies.

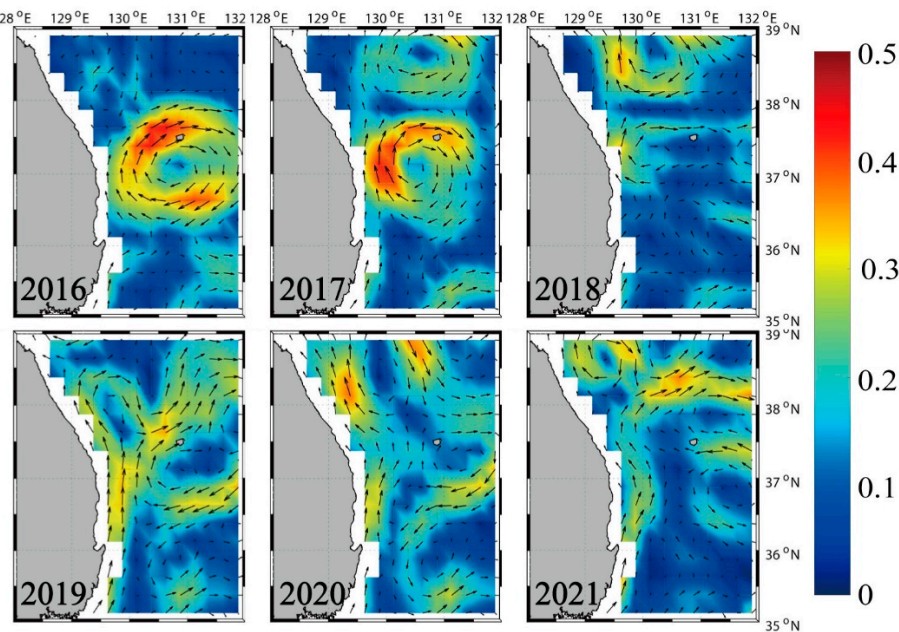

**Figure 5.** Inter-annual changes in surface geostrophic current (m/s) in the western part of East/Japan Sea (WES).

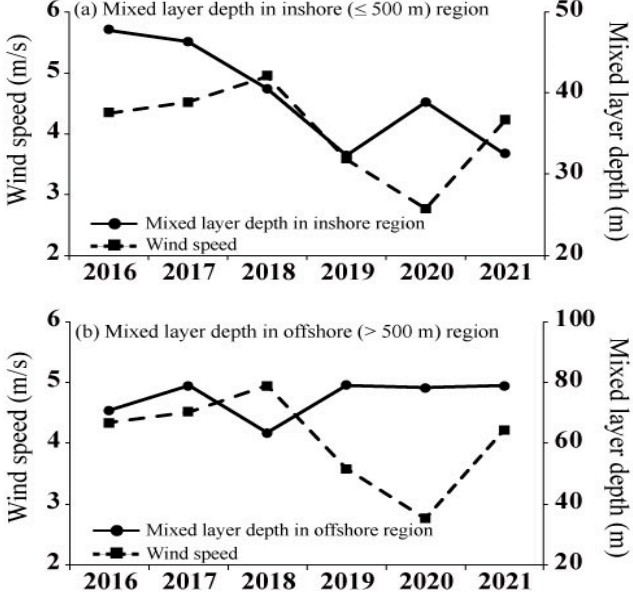

**Figure 6.** Inter-annual changes in the averaged mixed layer depth (MLD) of (**a**) offshore and (**b**) in-shore regions, along with wind speed in the western part of East/Japan Sea (WES).

### 3.3. Recent Oceanic Conditions in the Western Part of East/Japan Sea: Spatial Distribution of the East Korea Warm Current

The horizontal distribution of the water mass indicating the EKWC was analyzed using water temperature and salinity at the depth of 100 m. In 2016 and 2017, the central location of the EKWC was located around 35.6° N, and in 2018 and 2019, the central part of the EKWC was located near latitude 35.2° N, which was lower than that of the previous two years (Figure 7a). In 2020, water mass indicating the EKWC did not appear at

100 m, and, in 2021, the central part of the EKWC was distributed at the highest latitude (35.8° N) during the analysis period (Figure 7a). The inter-annual changes in the vertical length of the water mass indicating the EKWC from 2016 to 2018 was 82–89 m (Figure 7b). However, it gradually decreased after 2018, with the lowest level of 31.8 m recorded in 2020 (Figure 7b), and then, it increased again in 2021, with a vertical length of 79.2 m (Figure 7b). Analysis of the horizontal and vertical distributions of the EKWC showed clear inter-annual fluctuations. In 2017 and 2021, the horizontal and vertical range of the EKWC expanded, and the inflow of warm and saline water mass into the WES increased. In 2018, 2019, and 2020, the horizontal and vertical distribution ranges of the EKWC were reduced, and the influence of warm and saline water mass flowing into the WES decreased.

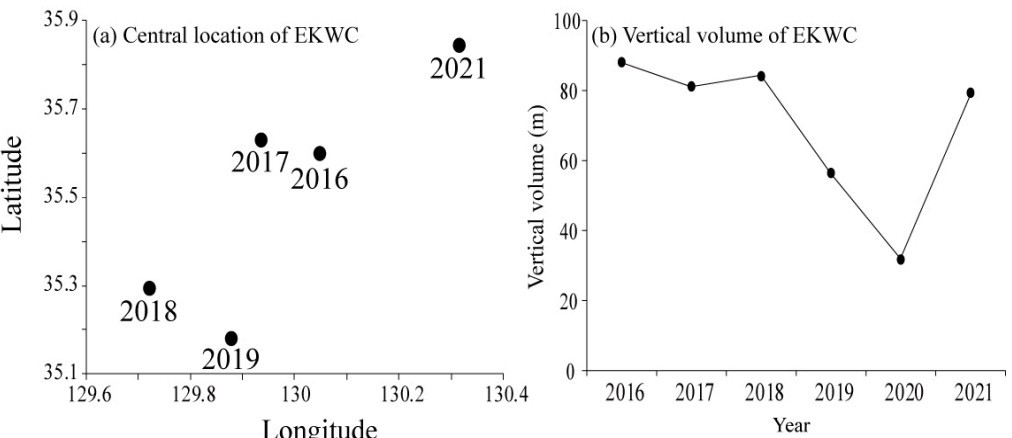

**Figure 7.** Inter-annual changes in (**a**) central location and (**b**) vertical volume of East Korea Warm Current (EKWC).

*3.4. Inter Annual Changes in Climate Index and Atmospheric Conditions in the Western Part of East/Japan Sea*

The intensities of AL, SH, and EAWM, which affect the changes in the atmospheric conditions around the Korean Peninsula during winter, showed clear annual changes. In particular, the interaction between SH and AL acted as a major factor influencing for wind speed and air temperature in the WES (Figures 8 and 9). ALPI was strong and positive in 2016, and it was negative from 2017 to 2021 (Figures 8 and 9). In particular, a strong low pressure was maintained in both January and February 2016 (Figures 8 and 9). However, in 2017 and 2020, a weak low-pressure was maintained in January and February. In 2018, 2019, and 2021, the intensity of AL strengthened during January, and it sharply weakened during February (Figures 8 and 9). The SH index was strongly positive in 2016 and 2018; however, it was negative in 2017, 2019, 2020, and 2021 (Figures 8 and 9). In 2016 and 2018, the SH strengthened in January and remained continuously strong until February; however, it continuously weakened from January to February in 2017, 2019, and 2020 (Figures 8 and 9). In 2021, the intensity of SH strengthened in January, but the strength of the SH decreased sharply in February (Figures 8 and 9). EAWMI showed positive indexes in 2016 and 2018 (Figures 8 and 9). However, the intensities of SH and AL in the two periods showed different fluctuation patterns. In 2016, the intensities of SH and AL were strong at the same time, and as a result, the pressure gradient between the two types of air masses was stronger, and therefore, the intensity of the northwest wind passing into the Korean Peninsula was strong. In 2018, the pressure gradient between the SH and AL was strong; additionally, the intensity of the northwest wind speed was strong, but the change in the intensity of the SH (rather than the AL) was the main factor controlling the intensity of the monsoon (Figures 8 and 9). However, in 2017, 2019, 2020, and 2021, when the intensities of SH and AL were weakened simultaneously, the pressure gradient between the two types of air masses was weakened, and the EAWM also showed a negative phase (Figures 8 and 9).

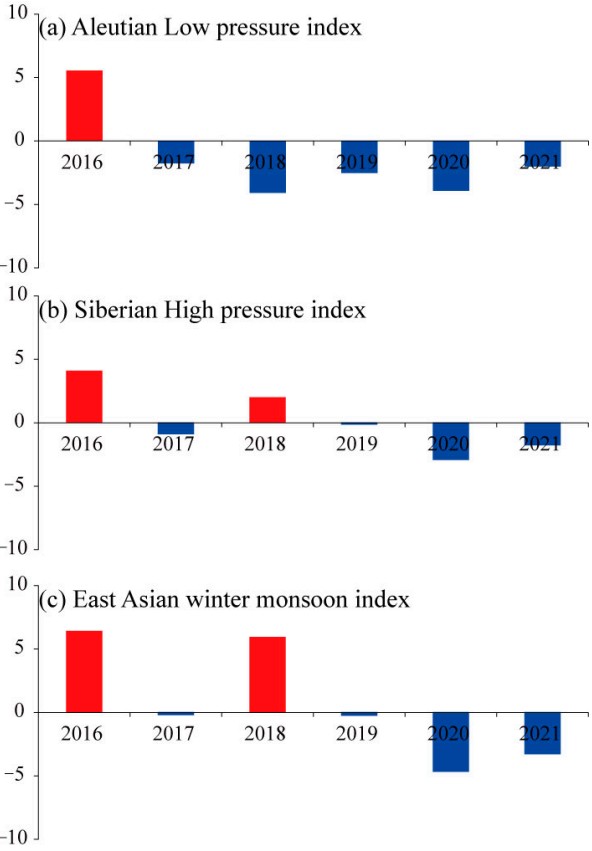

**Figure 8.** Inter-annual changes in the climate indexes: (**a**) Aleutian low pressure, (**b**) Siberian high pressure, and (**c**) East Asian winter monsoon index.

We observed that the inter-annual changes in atmospheric conditions, such as wind speed and air temperature in the WES, were similar to the annual variations in the intensity of air masses such as SH and AL. The inter-annual changes in air temperature in the WES had similar trends to the changes in the intensities of the SH and East Asia winter monsoon. In particular, in 2016 and 2018, the air temperature around the WES decreased, while the SH and winter monsoon became stronger. Conversely, in 2017, 2019, 2020, and 2021, the air temperature around the WES increased with the weakening of the SH and winter monsoon (Figures 10 and 11).

The inter-annual change in wind speed around the WES was similar to the inter-annual change in the pressure gradient between the SH and AL, and this pressure gradient was affected by the intensity changes in the SH and AL (Figure 11a). In particular, in 2016, when both the SH and AL were strengthened, the pressure gradient between the SH and AL increased, resulting in an increase in the wind speed in the WES (Figure 11a). Conversely, in 2020, the intensity of both the SH and AL weakened, resulting in the weakest pressure gradient between the SH and AL, resulting in the lowest wind speed in the WES. However, in some periods, the inter-annual change showed the opposite trend between the wind speed in the WES and the pressure gradient between the SH and AL as shown in Figure 11a.

The annual change in the wind speed around the WES has a stronger relationship with the pressure gradient between the SH and KE (Figure 11b) than the pressure gradient between the SH and AL (Figure 11a,b). Inter-annual changes in SLP in KE indicated annual fluctuation characteristics that were more comparable to the intensity of AL than that to spatial shift of AL (Figure 11c). In particular, in 2017, the central pressure of the AL weakened, but the central part of AL moved further southward. As a result, the SLP in KE decreased, the pressure gradient between the SH and SLP in the KE increased, and the wind speed around the WES also increased (Figure 11c). As such, inter-annual changes in wind

speed around the WES were affected by changes in the intensity and spatial distribution of the SH and AL simultaneously (Figure 11).

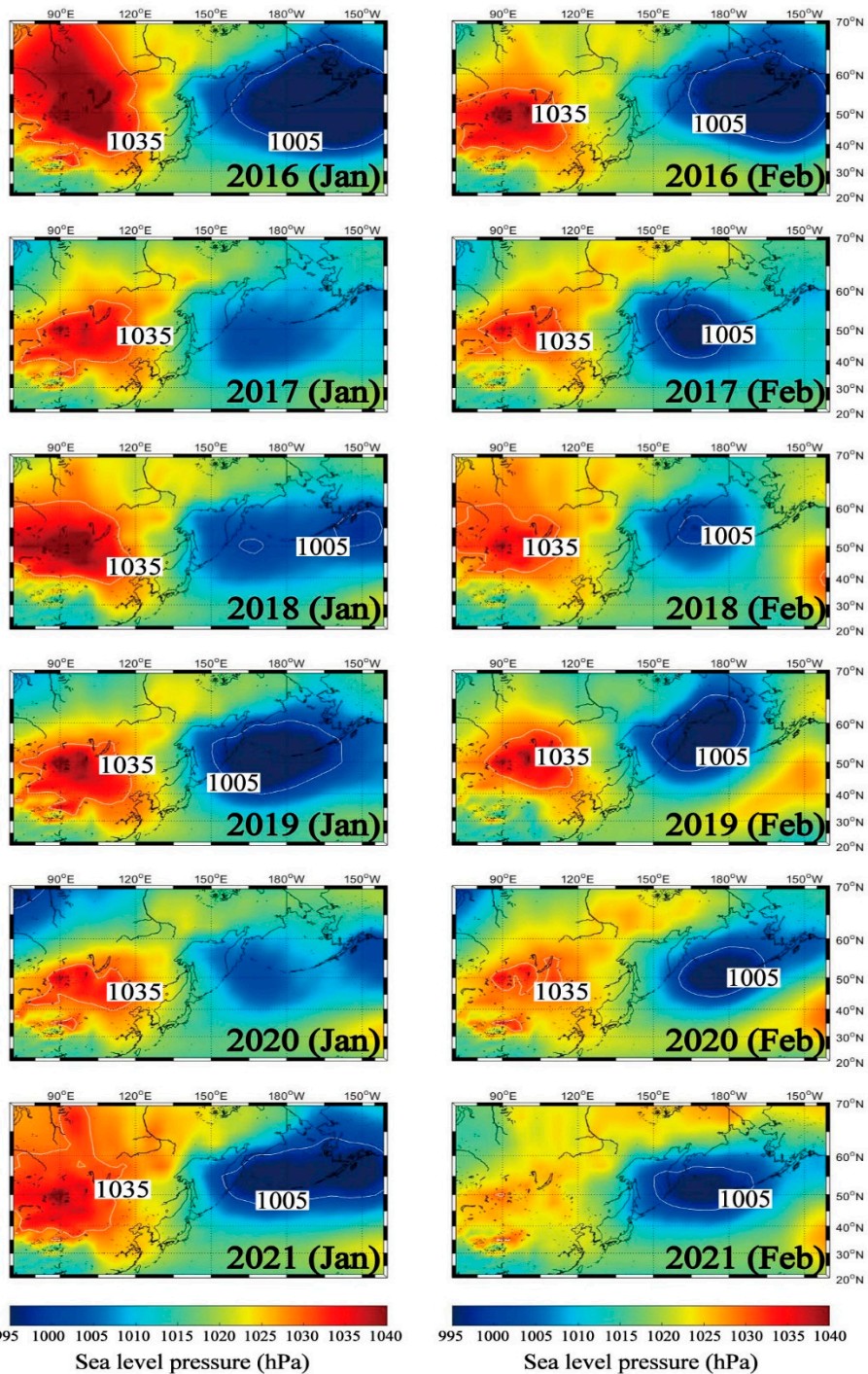

**Figure 9.** Inter-annual changes in climate index during the months of January and February for the entire time series of 2016–2021.

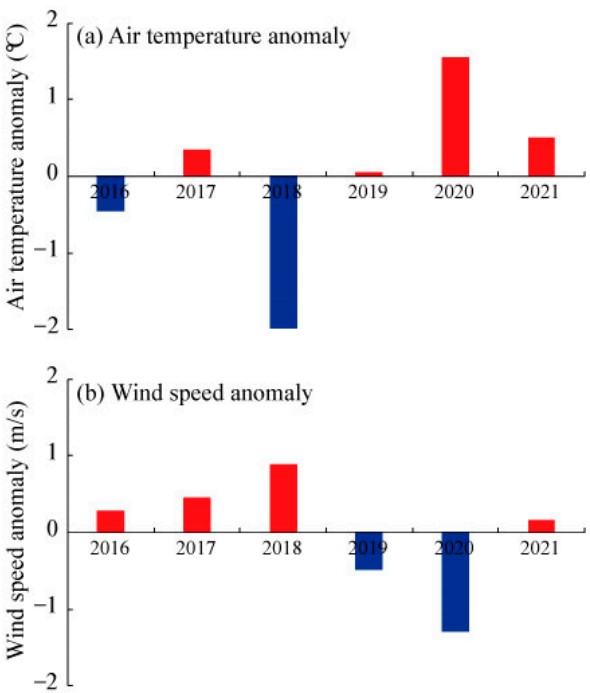

**Figure 10.** Inter-annual changes in (**a**) air temperature and (**b**) wind speed anomalies in the western part of East/Japan Sea.

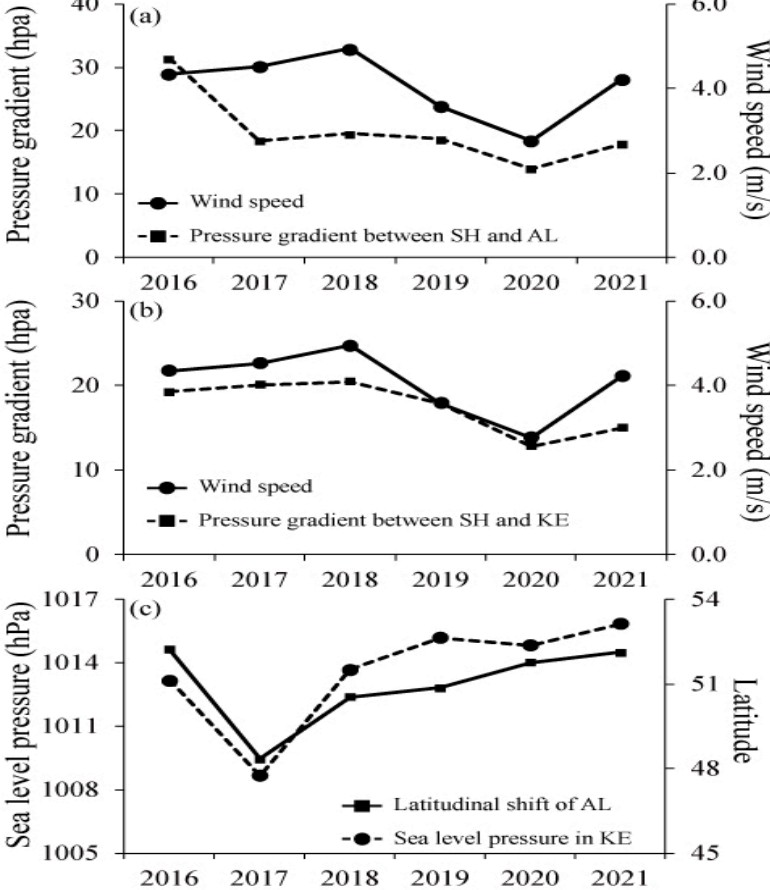

**Figure 11.** Inter-annual changes in wind speed in the western part of East/Japan Sea and pressure gradient between (**a**) Siberian high (SH) and Aleutian low (AL), (**b**) SH and Kuroshio Extension (KE) and (**c**) latitudinal shift of Aleutian low with sea level pressure in the KE region.

### 3.5. Inter Annual Changes in Oceanic Conditions in the Kuroshio Current Region in the East China Sea (ECS)

In the upper layer (≤200 m) of the PN line in winter (January), saline and warm water masses were continuously formed, water temperature was 22 °C or higher, and salinity was 34.75 or higher (Figure 12). However, changes in the spatial distribution of warm and saline water masses (temperature: ≥22 °C, salinity: ≥34.75 psu) located in the upper layer showed inter annual fluctuation clearly (Figure 12). In 2016, the western boundary of the warm and saline water in the upper layer was formed around 127° E. The warm and saline water mass in 2017 expanded further west and formed a western boundary around 126.4° E. After 2017, the western boundary moved eastward more consistently, and as a result, the western boundary of the warm and saline water masses in 2018 and 2019 formed around 127° E, as in 2016 (Figure 12).

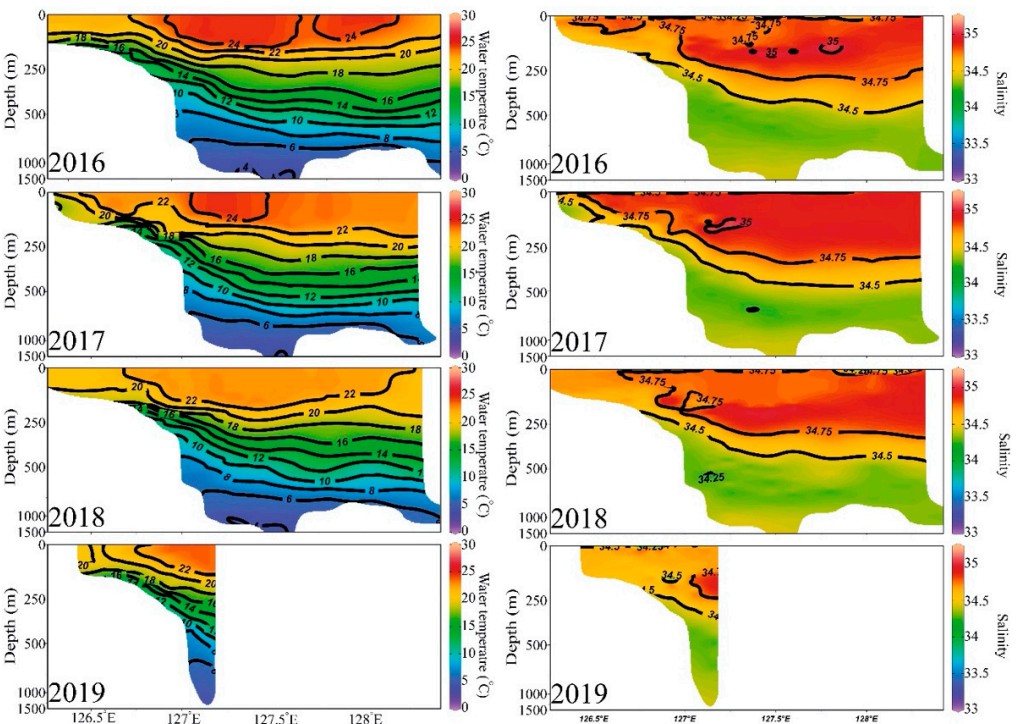

**Figure 12.** Inter annual changes in vertical distribution of water temperature and salinity along the PN line.

## 4. Discussion

The most recent CRS occurred in the late 1990s and it was characterized by spatial patterns in the sea surface temperature (SST), which had little resemblance to those associated with the dominant pattern of the variation in the North Pacific climate [13]. The CRS that occurred in the late 1990s showed more pronounced changes in spatial patterns, along with changes in the intensity [13–16], which is related to the second mode of SST in the North Pacific [13]. Long-term changes in the oceanic conditions of the WES have been affected by changes in climate factors, indicating atmospheric and oceanic conditions in the North Pacific [5,7]. After the CRS that occurred in the late 1980s, changes in the intensity of AL have acted as major factors that influence changes in the oceanic conditions of the WES [5,6]. However, in the late 1990s, the component of the changes in the central part of AL was more dominant than its intensity [13,29]. As a result, the mechanism of the CRS that occurred in the late 1990s influences the changes in the oceanic conditions of the WES, which may be different from the CRSs that occurred in the late 1980s. The recent oceanic conditions in the WES showed large inter-annual fluctuations, and it is related to atmospheric and oceanic circulations in the NPO based on the latitudinal shift in the central part of AL.

Changes in atmospheric conditions such as wind speed and air temperature were the main factors influencing changes in the oceanic conditions. The inter-annual changes in wind speed and air temperature around the WES were simultaneously affected by SH and AL [5,7]. After the CRS that occurred in the late 1980s, the change in the pressure gradient force between the central pressures of SH and AL affected the fluctuations in wind speed and air temperature around the WES [5,7]. However, after the CRS that occurred in the late 1990s, the recent changes in atmospheric conditions of the WES are more comparable with the pressure gradient between the SH and KE region than that between SH and AL. In particular, the change in the SLP over the KE region caused by the latitudinal shift of AL is a major factor that affects the atmospheric conditions in the WES. The most dominant variability of AL was the change in intensity with decadal-scale variability [41–43]. However, the changes in the latitudinal position of the central part of the AL are related to regional differences in oceanic and atmospheric conditions in the North Pacific [44,45]. The latitudinal position of AL variation corresponds well to the Western Pacific teleconnection pattern, which consists of a north–south dipole of sea level anomalies between the subtropical North Pacific and the Kamchatka Peninsula [44,45]. As a result, the spatial change in AL affects the winter wind speed, air temperature, and storm track in the Far East Asian and Western Pacific regions [45,46], and it has a significant correlation with the changes in the intensity of the winter monsoon flowing into East Asia, including the Korean Peninsula [47,48]. When the central part of the AL moved to the southern region, the SLP in the KE region was lower than normal, and the pressure gradient force between Siberia and the KE region was strengthened accordingly. As a result, the intensity of the winter monsoon flowing into the WES was strengthened and the air temperature in the WES was lowered. As such, the latitudinal shift of AL can be one of the important factors that controls the atmospheric conditions in the WES [5,7].

Changes in AL are related to the oceanic circulation system in the North Pacific. Additionally, changes in the intensity of the AL affect the changes in the intensity of the subtropical and subpolar gyres in the KE region [46,49]. In particular, after the occurrence of the CRS in the late 1980s, the strength of the Kuroshio Current rapidly weakened along with the sharp weakening of the AL; these changes in oceanic circulation in the NPO acted as a major factor influencing changes in the oceanic conditions of the WES [5,7]. However, the spatial shift of AL was also an important factor to changes in atmospheric oceanic conditions of the NPO. The latitudinal shift of AL affects the temporal changes in the boundary between the sub-arctic and sub-tropical gyre in the KE region. The latitudinal shift of AL corresponds to a change in latitudinal position of westerlies. During the north (south) shifted AL period, the latitudinal position of the westerlies was located at higher (lower) latitudes than normal, and the gyre boundary in the KE region moved northward (southward) [45]. As a result, the oceanic Rossby wave formed as a result of the baroclinic response to the AL movement that influenced the SST in the KE region [45], and the SLP and SST in the KE increased during the northern shift of the AL period [28,45,50,51].

Fluctuations in the oceanic conditions in the KE region associated with the intensity and spatial changes of AL were related to the volume transport of warm and saline water mass flowing in the WES. In particular, the change in the intensity of the Kuroshio Current associated with the change in the intensity and spatial location of the AL acted as a major factor influencing the change in the volume transport of warm and saline water mass flowing into the WES. The intensity of the Kuroshio Current in the ECS was related to the intensity and direction of the Kuroshio branch current, such as TWC and EKWC, which was associated with the changes in the main path of the Kuroshio Current [5,22]. When the intensity of the Kuroshio Current strengthened, the water temperature and salinity in the upper layer around the Ryukyu Islands in the East China Sea increased, and the main path of the Kuroshio Current moved more eastward by the enhanced Ekman transport [29,52]. As a result, the volume transport of warm and saline water mass separated from the Kuroshio Current and flowing into the WES decreased, and the water temperature in the upper layer and seal level in the WES were lower than normal. However, the water

temperature and sea level in the KE region increased [5,22]. Recent changes in oceanic conditions in the WES were also affected by the fluctuation patterns of the Kuroshio Current. When the warm and saline water mass observed in the PN line moved further westward (Weak Kuroshio), the horizontal and vertical ranges of warm and saline water mass flowing into the WES (EKWC) increased. Conversely, when the main path of the Kuroshio moved to the east (strong Kuroshio), the distribution range of the EKWC in the WES reduced.

After the late 1990s, the recent changes in the oceanic conditions of the WES have clearly shown inter-annual changes under spatial and intensity changes of climate factors. However, regional responses of oceanic conditions in the WES to the variations in climatic factors showed differences. The southern part of the WES was continuously affected by the EKWC, and the annual fluctuation of the oceanic conditions in the upper layer was not as large as that observed in other regions. Notably, the range of fluctuations in oceanic conditions related to changes in the pathway and intensity of the EKWC increased with increasing latitude [1,35]. Recent conditions in the WES also showed similar inter-annual changes, unlike those in the southern part of the WES, where the water temperature was continuously maintained at 13 °C or higher; the range of inter-annual fluctuations increased with increasing latitude. The inter-annual changes in the oceanic conditions in the WES clearly showed both horizontal and vertical differences. In particular, when the water temperature in the upper layer of the inshore region increased, the water temperature below the upper layer showed a decreasing pattern. The different fluctuation patterns between the upper and bottom layers in the inshore region were related to the vertical transfer of heat energy [36]. The stratification formed between the upper and bottom layers was strengthened with increasing water temperature in the upper layer, and as a result, the effect of transferring heat energy to the bottom layer (through the advection of solar radiation and turbulence) reduced. As a result, the water temperature in the upper layer increased continuously. However, the water temperature in the bottom layer indicated a decreasing pattern because of the decreased heat energy transferred from the upper layer to the bottom layer [35,36]. The oceanic conditions in the WES were simultaneously affected by different water masses, which have different mechanisms of formation and different origins. As a result, fluctuations in the oceanic conditions in each depth layer in the WES can indicate different fluctuation patterns. In particular, the fluctuation of the upper layer in the inshore region was mainly affected by the warm water mass originating from the equatorial current system flowing into the WES through the ECS [1,2,5]. On the other hand, oceanic conditions under the thermocline were affected by the cold-water mass originating from the coastal area of Vladivostok in Russia, passing into the coastal area of the WES [53,54]. Changes in atmospheric conditions in the coastal areas of Vladivostok and the ECS were affected by global atmospheric circulation, such as the AO, but showed contradictory fluctuations [55,56]. These changes in atmospheric conditions also affected the oceanic conditions, and the changes in the cold and warm water masses formed in the coastal areas of Vladivostok and East China Sea showed different patterns and flowed into the coastal area of the WES [55,56]. This variation can be a major cause for the different fluctuation patterns of oceanic conditions in the upper layer and below the thermocline. The oceanic mixed layer is induced by both atmospheric and oceanic forcing, such as wind stirring, internal waves, and advection [3]. The distribution and variability of the MLD in the WES are affected by geographic location and the high spatiotemporal variability of atmospheric forcing [57,58]. However, the main factors affecting the change in the MLD according to the geographical and physical conditions are different for each region, and they are divided into regions mainly affected by air–sea interactions, such as buoyancy flux and wind stress [57,58], and oceanic conditions, such as lateral advection [59]. In the WES, the recent condition of the MLD in the inshore region was mainly affected by changes in atmospheric conditions, such as wind speed and intensity of stratification due to the increased water temperature in the upper layer; conversely, the fluctuations of MLD in

the offshore region were mainly affected by changes in oceanographic conditions, such as eddy activity.

Our study provides important insights into the recent oceanic conditions in the WES associated with recent climate change, and explores the main climate factors that influence the changes in the oceanic conditions in the WES. After recent CRS, the component of spatial change in climate factors was more dominant than changes in intensity, and thus, climate change after the late 1990s influenced changes in the oceanic conditions of the WES through different mechanisms and processes from the previous CRS. In particular, after the late 1980s CRS, changes in the intensity of AL acted as major factors affecting the changes in oceanic conditions of the WES. However, after the CRS that occurred in the late 1990s, recent changes in the oceanic conditions of the WES were more influenced by the spatial changes in the central part of AL than that by intensity. In addition, the response of oceanic conditions to climate change was different for each region and depth layer. In this study, we obtained a better understanding of the responses and mechanisms of the fluctuation of oceanic conditions in the WES with respect to climate change and this can contribute to research aiming at the prediction of future oceanic conditions. In a further study, based on these results, it will be necessary to investigate and understand the key factors that affect the changes in the oceanic conditions of the WES by each CRS period through long-term analysis in oceanic conditions of the WES and climate variability in the NPO.

**Author Contributions:** H.-K.J.; conceptualization, methodology, validation, formal analysis and writing—original draft, C.-I.L.; supervision, project administration and funding acquisition, S.M.M.R. and H.-C.C.; data curation and visualization, J.-M.P.; writing—review and editing. All authors have read and agreed to the published version of the manuscript.

**Funding:** This research was supported by the National Institute of Fisheries Sciences R&D project (R2021032). This research was a part of the project entitled "Long-term change of structure and function in marine ecosystems of Korea" funded by the Ministry of Oceans and Fisheries, Korea. This research was also supported by the National Research Foundation of Korea (NRF) grant funded by the Korea government (MSIT) (No. 2020R1F1A1051773).

**Institutional Review Board Statement:** Not applicable.

**Informed Consent Statement:** Not applicable.

**Data Availability Statement:** Not applicable.

**Acknowledgments:** We would like to thank the members of the Fisheries Resources and Environment Research Division for their assistance in field observations and data analyses.

**Conflicts of Interest:** The authors declare no conflict of interest.

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
