# Peer review of "Recent Trends in Oceanic Conditions in the Western Part of East/Japan Sea: An Analysis of Climate Regime Shift That Occurred after the Late 1990s"

_jmse, doi:10.3390/jmse9111225_

Round 1

Reviewer 1 Report

There is a good story here and the regional variability is interesting. I believe that the manuscript would be more readily received by the scientific community if the clarity of the written expression is improved. Overall, I believe that the paper would benefit from a minor revision. The following is my suggestion.

  1. The line 23-26 is too long for a paragraph. It is better to rephrase it.
  2. Line 141: it should be Determination of the mixed layer depth (MLD)
  3. Likewise, line 152: Determination of geographic location...
  4. Line 157: correction ..., we selected a region of 36.5° N or...
  5. Line 165-170: the authors should provide a link of the data. 

Author Response

We thank the reviewer for the detailed review of our manuscript and are grateful for the valuable suggestions and encouragement. We have improved upon the quality of the writing and improved the overall readability by engaging a professional editing service. We are pleased to state that all the suggested modifications were carried out in the revised manuscript.

Reviewer 2 Report

The East/Japan Sea (WES) is an important area to explore the climate change, atmospheric and oceanic conditions, and this paper analyzed the temperature and salinity using CTD data. Some interesting phenomenon was found, such as the correlation among the temperature variation of different depth, the relation between wind speed and MLD, and relation between MLD and currents (eddy). But the weakness is the short time series data (2016-2021) can not enough explain the recent oceanic condition trends in the WES after the recent climate regime shift (CRS) in the North Pacific (occurred in the late 1990s). More evidence needs to be found to support the author's point of view that the component of spatial change in climate factors was more dominant than changes in intensity, and thus, climate change after the late 1990s influenced changes in the oceanic conditions of the WES through different mechanisms and processes from the previous CRS. Writing needs to be strengthened. Figures is too much and need to be merged, such as figures 3-6, figures 10-12, figures 16-17. 

Comments are as follows

Introduction

  1. In second paragraph, the Tsushima Current should be drawnin the figure, so that the reader could clearly understand where the Tsushima Current comes from and through where it enters the East/Japan Sea.
  2. Both in the second paragraph and five paragraph describedthat the three CRSs may caused by different mechanisms, the spatial patterns or the intensity.

Data and methods

  1. In 2.1.3, “Thecentral location of the EKWC was calculated using the average latitude and longitude of the regions”. Please explain it in detail.

Results

The figure should be merged.

Discussion

General, most of the discussions are quoting the opinions of others, and the author’s conclusions are not marked from which figure.

Author Response

We thank the reviewer for the detailed review of our manuscript and are grateful for the valuable suggestions and encouragement. We have improved upon the quality of the writing and improved the overall readability by engaging a professional editing service. We are pleased to state that all the suggested modifications were carried out in the revised manuscript.

The purpose of this study is to explain the pattern of fluctuations in recent oceanic conditions of the Western part of East/Japan Sea and its possible mechanisms. To understand such climate change phenomena, we compared climate regime shift (CRS) that occurred after the late 1980s and late 1990s. We have explained the mechanisms that influence the dominant components such as intensity and spatial change in the two different periods that affect changes in the oceanic conditions in the Western part of East/Japan Sea. We do understand and agree with your concern about the short time series data, which perhaps is not adequate to explain the recent trends in oceanic conditions in the WES after the recent climate regime shift in the North Pacific. However, additional evidence has been added in the manuscript to add credence to our study. 

We thank you for your nuanced suggestion, which has helped us in improving the quality of this manuscript.
